# Influence of Metallic Oxide Nanoparticles on the Mechanical Properties of an A-TIG Welded 304L Austenitic Stainless Steel

**DOI:** 10.3390/ma13204513

**Published:** 2020-10-12

**Authors:** Sebastian Balos, Miroslav Dramicanin, Petar Janjatovic, Nenad Kulundzic, Ivan Zabunov, Branka Pilic, Damjan Klobčar

**Affiliations:** 1Faculty of Technical Sciences, University of Novi Sad, Trg Dositeja Obradovica 6, 21000 Novi Sad, Serbia; dramicanin@uns.ac.rs (M.D.); janjatovic@uns.ac.rs (P.J.); kulundzic@uns.ac.rs (N.K.); 2Faculty of Special Technology, Alexander Dubček University of Trenčín, Študentská 2, 911 50 Trenčín, Slovakia; ivan.zabunov@tnuni.sk; 3Faculty of Technology, University of Novi Sad, Bulevar Cara Lazara 1, 21000 Novi Sad, Serbia; brapi@uns.ac.rs; 4Faculty of Mechanical Engineering, University of Ljubljana, Aškerčeva c. 6, 1000 Ljubljana, Slovenia

**Keywords:** oxide flux, activated tungsten inert gas welding, mechanical properties, 304L stainless steel

## Abstract

Austenitic stainless steels represent a significant aerospace material, being used for various castings, structural components, landing gear components, afterburners, exhaust components, engine parts, and fuel tanks. The most common joining process is tungsten inert gas (TIG) welding, which possesses many advantages such as suitability to weld a wide range of ferrous and non-ferrous metals and alloys, providing high quality welds with good mechanical properties. Its major disadvantage is low productivity due to low penetration and welding speed. This can be overcome by introducing an activating flux before welding. The activating flux reverses the material flow of the weld pool, significantly increasing penetration. Therefore, shielding gas consumption is reduced and welding without a consumable is enabled. However, the consumable in conventional TIG also enables the conditioning of the mechanical properties of welds. In this study, Si and Ti metallic oxide nanoparticles were used to increase the weld penetration depth, while bend testing, tensile, and impact toughness were determined to evaluate the mechanical properties of welds. Furthermore, optical emission spectroscopy, light, and scanning electron microscope were used to determine the chemical compositions and microstructures of the welds. Chemical compositions and weld mechanical properties were similar in all specimens. The highest tensile and impact properties were obtained with the specimen welded with the flux containing 20% TiO_2_ and 80% SiO_2_ nanoparticles. Although lower than those of the base metal, they were well within the nominal base metal mechanical properties.

## 1. Introduction

Austenitic stainless steels are used in various applications due to their excellent corrosion resistance in atmospheric and pure water environments, marine environments as well as in acid and alkaline solutions. Their advantages also present their unique properties such as hardness, strength, density, malleability, ductility at lower temperatures, elasticity, conductivity, thermal expansion, good heat, and fire resistance. Austenitic stainless steels are also easily fabricated using forming, machining, cutting, and welding, which makes them good materials for producing components in the aerospace industry. Common applications of austenitic stainless steels in aerospace applications include various castings, structural components, landing gear components, afterburners, exhaust components, engine parts, and fuel tanks. Depending on the use and desired properties of the components, the following austenitic stainless steels are usually used in aerospace applications: 304, 316, and 321 [1,2,3].

The dominant welding process common for welding austenitic stainless steels is tungsten inert arc (TIG) welding. To increase productivity, the activated flux can be used prior to TIG welding, resulting in the modified process called activated tungsten inert gas welding (A-TIG), which has significantly increased penetration that is sufficient for full penetration of the whole plate thickness. Therefore, in A-TIG, common V-preparation can be replaced by a simpler and cheaper closed square preparation of the base metal, since the whole contact surface between base metals can be welded in one pass. This renders multi-pass welding with repeated welding of incremental welds with consumable material unnecessary. Consumables are not needed, a considerably lower amount of shielding gas is expended, and less power and time is needed. A-TIG has been demonstrated to be effective in welding various base metals such as different types of steels, titanium alloys, aluminum alloys, magnesium alloys, and dissimilar materials [4,5]. All these are accompanied by a considerable energy consumption and a lower man-power demand [6,7].

A-TIG was invented in the former USSR (Union of Soviet Socialist Republics), more accurately, at the Paton Welding Institute in Kiev, present-day Ukraine [8]. It usually consists of metallic oxides (SiO_2_, TiO_2_, Al_2_O_3_, NiO, MnO_3_, Cr_2_O_3_, CeO_2_, and V_2_O_5_) in alcohol or acetone solvents [5,9,10].

Aside from the flux particle type, the particle size is also influential, with a preference for nanoparticles. Several studies have proven that nano-sized particles possess a high efficiency. This is because the high specific surface area of the nano-sized particles results in a faster thermal dissociation and decomposition during arc heating compared to that of the micron-sized particles [11,12,13]. The flux is applied on the surface by a brush or in the form of a spray prior to welding. There are two major mechanisms of increasing the penetration: the reversal of molten metal flow and arc constriction. In conventional TIG welding, the molten metal flows from the center of the weld (low surface tension) toward the edges (high surface tension area), creating a relatively wide but shallow weld. In A-TIG welding, this is reversed due to the influence of oxygen from the flux, resulting in a completely different weld shape: narrow and deep [14,15]. The arc constriction is the result of the electronegativity of the metallic constituents in the flux [16,17]. A-TIG has been successfully used for welding a number of materials, ferrous and non-ferrous as well as dissimilar materials [18,19].

Although A-TIG possesses many advantages, the lack of a consumable may affect the mechanical properties of the weld itself. Namely, the consumable material in conventional TIG in overmatching welds has a higher ultimate tensile strength (UTS) compared to the base metal, thus influencing the rise in the UTS of the weld itself. However, there have been few studies addressing this important issue with A-TIG.

Sharma and Dwivedi [20] experimented with the dissimilar welding of P92 and 304H steels, where multipass TIG (with Inconel 82 wire consumable) and A-TIG (without consumable) were compared. It was found that the weld metal obtained by A-TIG contained untempered martensite, compared to austenite in the weld metal obtained by TIG. This resulted in a higher strength, but a lower impact toughness of the weld metal in A-TIG compared to TIG. In A-TIG welded specimens, tensile testing resulted in the fracture that occurred in the 304H base metal, while in TIG, the fracture occurred in the weld metal. Ramkumar et al. [21] studied the influence of activated flux in forming dissimilar welds between 316L austenitic stainless steel and S32750 super-duplex stainless steel. It was shown that the tensile fracture occurred in 316L steel and not in the weld metal or stronger S32750 steel. Hdhibi et al. [22] experimented with the remelting of 316L steel with TIG and various compositions of activated flux. The remelted specimens were tensile tested and it was determined that δ-ferrite plays a major role in increasing the tensile properties of predominantly austenitic material. The influence of nitrogen content in shielding gas (Ar based) composition was studied in [23] where it was found that the nitrogen content had a positive effect on the tensile properties and hardness of 304 austenitic stainless-steel A-TIG welds due to interstitial solid solution strengthening and grain refinement.

Instead of aiming at optimizing the flux to achieve the maximum penetration as in our previous study [12], in this study, the influence of nanoparticle-based fluxes on the mechanical properties of the weld was determined. The obtained mechanical properties were compared to the base metal and procedures for determining the weld quality, which may be the impeding factor of the wider acceptance of A-TIG technology in the industrial sector. The main novelty identified was finding the effect of nanoparticles in the flux and their effect on the weld shape and mechanical properties of welds.

## 2. Materials and Methods

In this study, the base metal used was AISI 304L austenitic stainless steel. Its chemical composition is presented in Table 1, while its mechanical properties are shown in Table 2. The base material size was cut from the 10 mm plate to 200 mm × 50 mm dimensions by waterjet, and then butt welds were done without a gap.

Prior to welding, several nanoparticle-based fluxes in acetone solvents were prepared. The active element of the fluxes was 20 nm TiO_2_ and SiO_2_ nanoparticles, weighed by a Tehtnica Type 2615 (Zelezniki, Slovenia) analytic balance, mixed with the solvent by means of a Tehtnica mm530 (Zelezniki, Slovenia) magnetic stirrer for 10 min. After mixing, the fluxes were smeared over the surface of the specimens to be welded by a brush, having the width of 20 mm at both base metal sides and were 10 µm thick, measured immediately after application. Subsequently, the welding was performed by using an EWM Tetrix300 TIG welding machine (EWM, Mündersbach, Germany). Welding current was 250 A and the WTh2 electrode tip was sharpened to 90°. The distance from the tip to the base metal was 2 mm, while the welding speed was set and kept constant at 40 mm/min. All specimens were welded without a consumable and with Ar shielding gas at a 14 L/min flow rate as well as a ceramic backing plate. The welded specimens were designated in relation to the flux applied during welding: 5Ti (100% TiO_2_), 4Ti1Si (80% TiO_2_; 20% SiO_2_), 3Ti2Si (60% TiO_2_; 40% SiO_2_), 2Ti3Si (40% TiO_2_; 60% SiO_2_), 1Ti4Si (20% TiO_2_; 80% SiO_2_), and 5Si (100% SiO_2_).

Weld characterization was done in macro imagery, microstructures, microhardness and tensile, bend, and impact testing. Tensile testing was conducted in accordance to EN ISO 4136:2012 standard, impact energy in accordance to EN ISO 9016:2012, bend testing with EN ISO5173:2009, and microhardness with EN ISO 6507:2018. Specimens were cut by a waterjet, in accordance with Figure 1. In Figure 1, the weld direction is indicated. Metallographic preparation was done on Struers equipment (Bellerup, Denmark) and consisted of cutting, mounting, abrasive paper grinding (150 to 2000 grit), polishing with diamond suspensions (6 to 0.25 µm particle size), and etching by using aqua regia (three parts 37% HCl; 1 part 70% HNO_3_). Examinations were done with a Leitz Orthoplan light microscope (Oberkochen, Germany) and JEOL JSM6460LV (Tokyo, Japan) scanning electron microscope (SEM). Microhardness was measured by a Wilson Tukon 1102 (Uzwil, Switzerland) device with 100 gf loading in the base metal and weld metal, as shown in Figure 2. The distance between the indentations was 0.5 mm. Weld metal chemical analysis was done by an ARL ISpark 8860 (Waltham, MA, USA) optical emission spectrometer (OES). Tensile and bend testing were done by the Schenck Hydropuls PSB 250 (Darmstadt, Germany) universal hydraulic testing machine, while impact testing was performed on a Testing Equipment IE JWT-450 (Jinan, China) instrumented Charpy pendulum tester. Image analyses to obtain the δ-ferrite content was done by ImageJ software (version 1.51).

## 3. Results and Discussion

### 3.1. Metallographic Testing and Chemical Compositions

Macro-sections of the welded specimens are shown in Figure 3. Although similar in shape and columnar morphology, typical for A-TIG weldments, there was a marked difference in width. In all specimens, the root was significantly wider than the face, as was obtained in [12], which was the result of the heat accumulation and the ceramic backing plate, which limited excessive penetration. This indicates that the maximum penetration with applied parameters can be higher, that is, the present parameters can be applied to weld a thicker base metal of the same type. Alternatively, this means that the same thickness as applied in this experiment could be welded with a lower welding current, or a higher welding speed. The widest weld was obtained in the 1Ti4Si specimen. However, it is difficult to assess the maximum penetration of tested fluxes with the parameters used in this experiment just by evaluating the weld metal shape from the macro-sections. Moreover, it is difficult to compare the present macro-sections to the macro-sections obtained by the same fluxes, because in [12], different welding parameters were used. At higher currents, larger particles may be optimal for achieving high penetration, but this remains to be proven in future studies.

In Table 3, the widths of the weld metals are shown. The addition of SiO_2_ nanoparticles makes the weld face narrower due to the arc constriction effect of the Si, with the only exception being 1Ti4Si. This is in contrast to the results shown in [12], where the 3Ti2Si specimen proved to have the highest penetration by using the same fluxes. However, it must be noted that the current used in this study was 300 A, which is considerably higher than that in [12], where it was 200 A. Obviously, 1Ti4Si represents the optimal mixture from the point of view of the welding parameters, most notably the welding current.

The microstructures obtained with the light microscope of specimen 5Ti is shown in Figure 4, where typical dendritic weld metal and transition weld interface microstructures were obtained. The SEM microstructure revealed the typical δ-ferrite morphology that occurs between austenitic dendrites. The SEM micrograph is shown in Figure 5, presenting δ-ferrite in the austenite matrix of the weld metal. There were no differences between the microstructures obtained with different experimental conditions. The δ-ferrite content in the base metal was 6%, while in the weld metal of all specimens, it was 7.63% to 8.01%.

Chemical compositions of weld metals obtained by OES as well as the 304L steel nominal chemical composition are shown in Table 4. It can be seen that all obtained chemical compositions of weld metals corresponded well to the nominal chemical composition of 304L austenitic stainless steel. Additionally, the weld metal chemical compositions corresponded well to the chemical composition of the base metal shown in Table 1, indicating that the flux does not influence changes in the chemical composition of the weld metal.

### 3.2. Mechanical Properties of Welds

The tensile properties of the welds are shown in Figure 6, while the typical fracture mode is shown in Figure 7. Tensile properties in all specimens were lower compared to the base metal, which could be the result of the replacement of polygonal austenitic by a cast dendritic microstructure, that is, a polygonal with dendritic austenite. Tensile strengths and reductions of area were well within the limits set up by ASTM A240/A240M relating to 304L austenitic stainless steel. The highest tensile strength and reduction in area of all welded specimens were obtained in the 1Ti4Si specimen, as the result of the widest weld metal and the widest space for deformation. This is also important from the point of view of penetration potential, since this mixture influences the largest weld metal. Fracture mode is fully ductile, with a notable necking, as shown in Figure 7. This was proven by the SEM image analysis (Figure 8). A dimpled specimen fracture surface was obtained, typical of ductile fracture. This is the result of microvoid formation around non-metallic inclusions, microvoid growth, and coalescence [24]. At the bottom of the dimples, circular phases can be observed, which can be morphologically attributed to non-metallic inclusions (Figure 8c,d).

The highest tensile strength is the result of the highest absolute amount of δ-ferrite in the austenitic matrix, which is proportional to the weld metal width, most particularly the weld metal width, as shown in Table 3. The body-centered cubic crystal lattice (BCC) had higher mechanical properties compared to the austenitic face-centered cubic lattice. This is similar to the findings of Hdhibi et al. [22], who also stressed the significance of δ-ferrite in obtaining the convenient tensile properties of 316L austenitic stainless steel A-TIG remelting. Furthermore, the tensile strengths obtained in this study were very similar to those obtained in Hdhibi et al. [22], who used 316L austenitic stainless steel welded/remelted by A-TIG with the flux based on SiO_2_ and TiO_2_. However, in this study, a positive correlation between SiO_2_ nanoparticle content and weld metal root width and tensile strength could be observed. Additionally, when single component fluxes are considered, it was found that the specimen welded with the flux containing SiO_2_ nanoparticles exhibited a higher tensile strength and a lower reduction in area compared to the specimen welded with the flux containing TiO_2_ nanoparticles.

Impact energies obtained by the instrumented Charpy pendulum tester, with V-notch situated in the center of the weld metal are shown in Figure 9. As in base metal (Table 2), crack propagation values were higher compared to the crack initiation energy values. However, all impact energy values were lower than those of the base metal. Values for overall impact energies were between 80% and 95% of the base metal. The highest overall impact energies were obtained in specimens 2Ti3Si and 1Ti4Si. The presence of the less ductile δ-ferrite might explain why weld metals have a slightly lower impact energies compared to the base metal. Figure 9a,b depicts the typical force–time curves from the instrumented Charpy pendulum tester of the base metal and 3Ti2Si specimen. It can be seen that the curve shape was very similar, however, the overall impact energy of the base metal specimen was higher (blue accumulative curve).

Although impact energies (crack initiation, propagation, and overall impact energies) were slightly lower in the welded specimens compared to the base metal, bend testing did not result in any cracks, on either side of any specimen tested (Figure 10). This is in accordance with the relatively high values in the reduction of area (Z) obtained in tensile testing (Figure 6). The deformed area of the weld metal, despite the fact that the surface was ground prior to testing, showed signs of columnar morphology, which is in accordance with the macro images (Figure 2). The values obtained in all specimens, with crack propagation energies higher than crack initiation energies, proved that the suitability of welds to pressure vessel application is maintained. Namely, a higher crack propagation energy ensures a minimized sudden failure of the weld in the case of overpressure occurrence.

The microhardness results are shown in Figure 11. In the majority of specimens, the hardness of the weld metal was higher compared to the hardness of fusion zones, probably due to a higher content of δ-ferrite and a lower distance between δ-ferrite phases: grain size in base metal 20–100 µm and interdendritic distance in weld metal 10–40 µm. This is depicted in Figure 4a,b. Although the differences were small, the microhardness of specimens 2Ti3Si and 1Ti4Si were the lowest (Figure 11c), which is interesting, since the tensile properties and impact energies of these specimens were the highest. Apparently, there was an optimal δ-ferrite content, which provided the highest mechanical properties.

## 4. Conclusions

According to the presented results and within the limitations of this study, the following conclusions can be drawn:Chemical compositions of the weld metals are not dependent on the composition of the flux.Different SiO_2_ to TiO_2_ ratios result in obtaining similarly shaped weld metals, with the same columnar structure, but with different widths.The widest weld resulted in the highest absolute amount of δ-ferrite, the widest space for deformation, resulting in the highest strength and ductility.Tensile properties and impact toughness of weldments were marginally lower compared to the base metal, due to the replacement of strain hardened polygonal austenite by the mixture of cast dendritic austenite.Tensile strength and reduction of area as well as impact toughness of the optimal specimen and the majority of other specimens, were within the limits set by the base metal standard limits.Bend testing did not result in the occurrence of any cracks.Microhardness values in the weld metal in the majority of specimens were higher compared to the fusion zones. Additionally, specimens that had the lowest average microhardness had the highest tensile and impact toughness.

## Figures and Tables

**Figure 1 materials-13-04513-f001:**
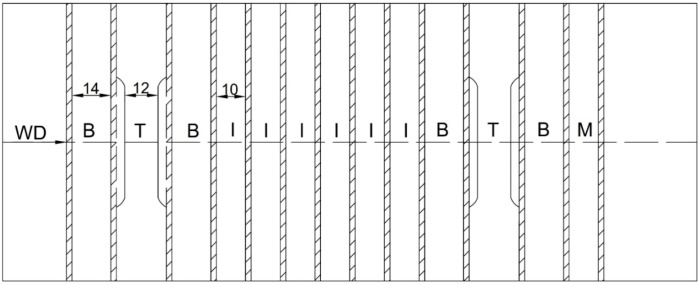
Specimens in a welded plate: M-macro; T-tensile testing; B-bend testing; I-impact testing, with indicated welding direction (WD) and specimen sizes, the thickness being kept as the plate itself. The dashed line indicates the position of the weld.

**Figure 2 materials-13-04513-f002:**
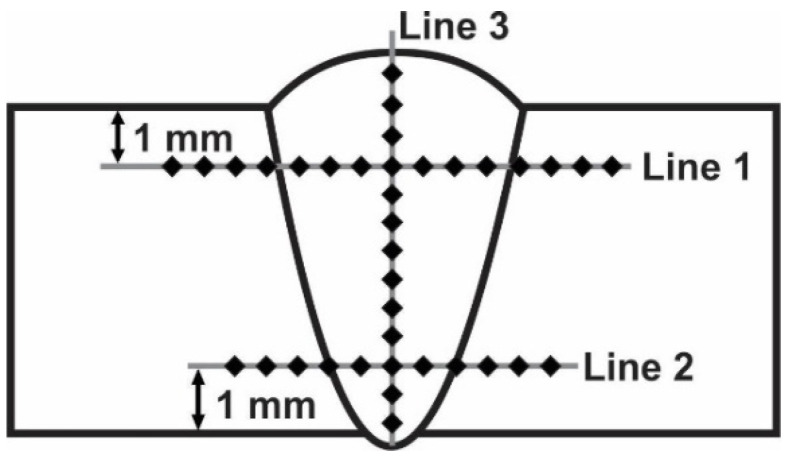
Microhardness measurement scheme.

**Figure 3 materials-13-04513-f003:**
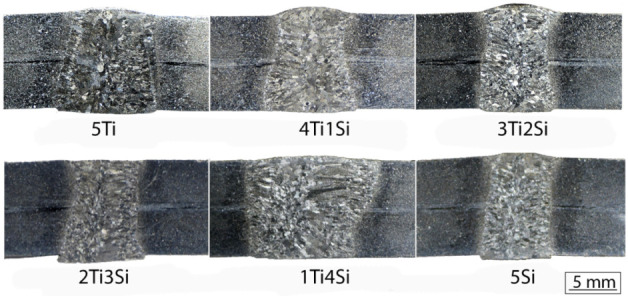
Macro-sections of specimens welded with various types of flux.

**Figure 4 materials-13-04513-f004:**
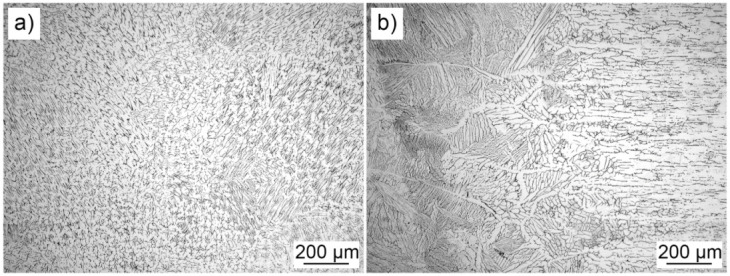
Microstructures of the 5Ti specimen obtained by light microscope: weld metal (**a**); weld interface (**b**).

**Figure 5 materials-13-04513-f005:**
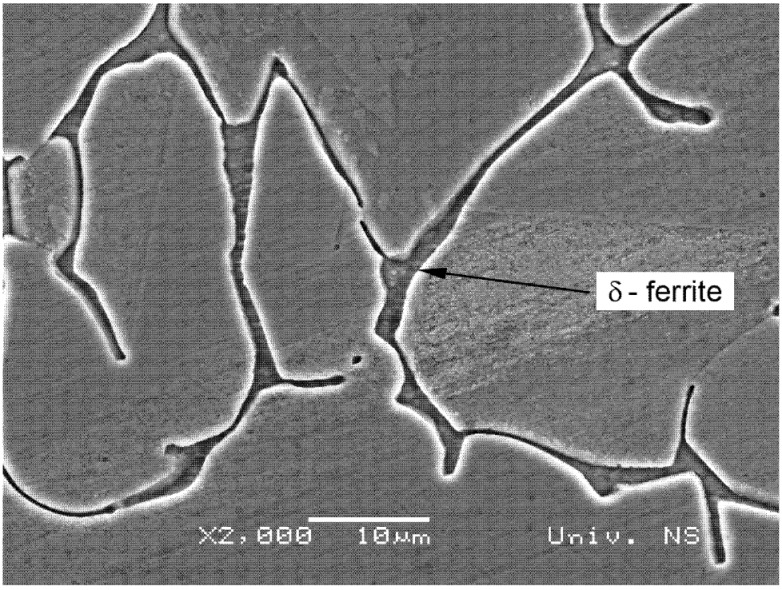
Scanning electron microscope (SEM) image of the δ-ferrite in the weld metal of the 5Ti specimen.

**Figure 6 materials-13-04513-f006:**
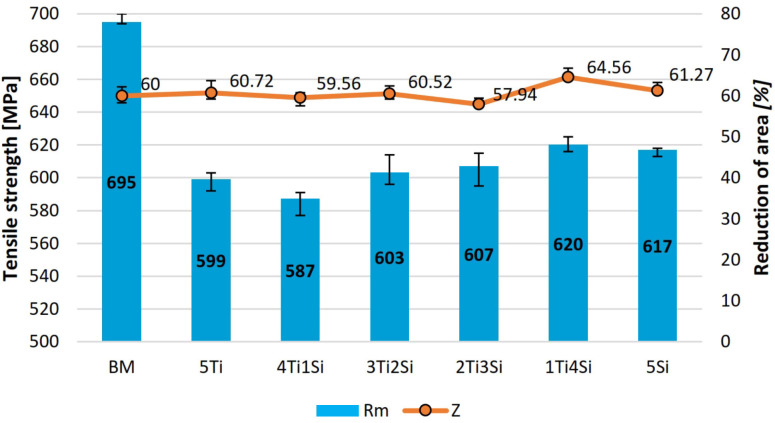
Tensile properties of the welds.

**Figure 7 materials-13-04513-f007:**
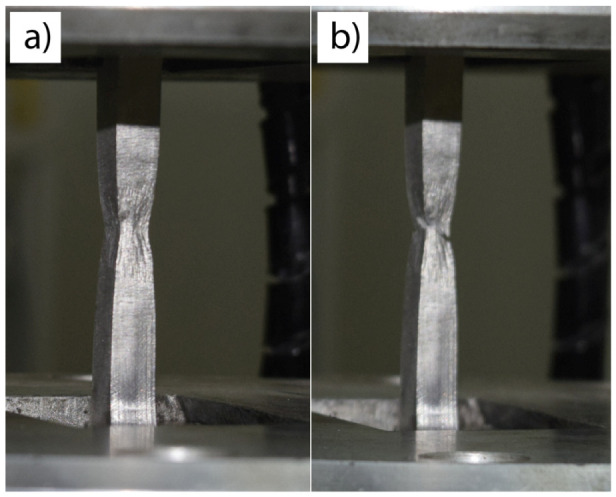
Necking in two tensile 1Ti4Si specimens: closer to the weld beginning (**a**); closer to the weld end (**b**).

**Figure 8 materials-13-04513-f008:**
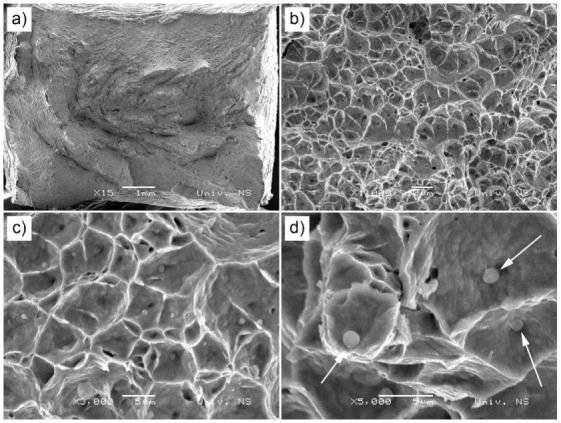
SEM images of fracture surface obtained by tensile testing of the 4Ti1Si specimen: macro image (**a**); micro images showing ductile fracture (**b**); dimples (**c**); dimples with indicated non-metallic inclusions (**d**).

**Figure 9 materials-13-04513-f009:**
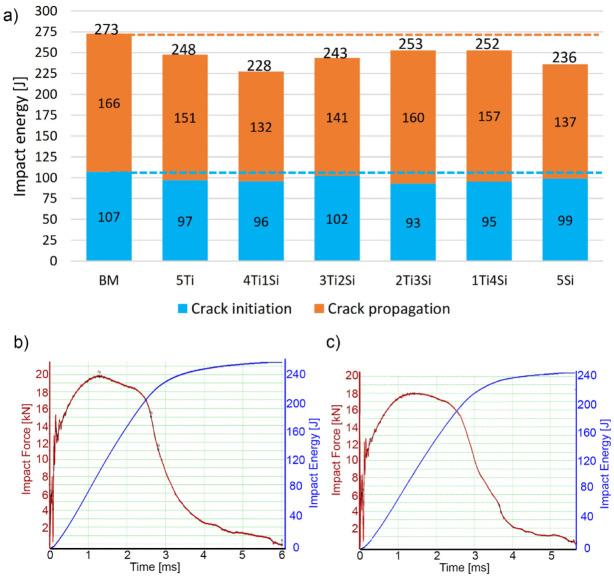
Impact energy of welded specimens: crack initiation, crack propagation, and overall impact energy (**a**); force–time curve of base metal (**b**); force–time curve of specimen 3Ti2Si (**c**).

**Figure 10 materials-13-04513-f010:**
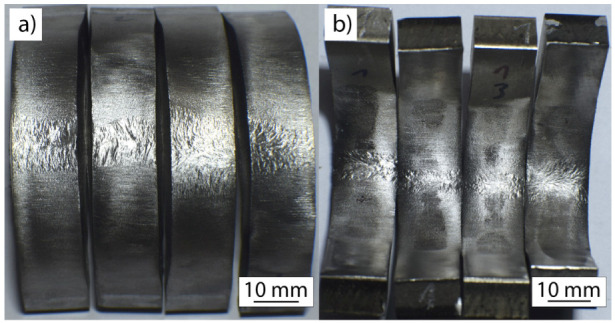
Bend testing of the 1Ti4Si specimen; weld root (**a**) and weld face (**b**).

**Figure 11 materials-13-04513-f011:**
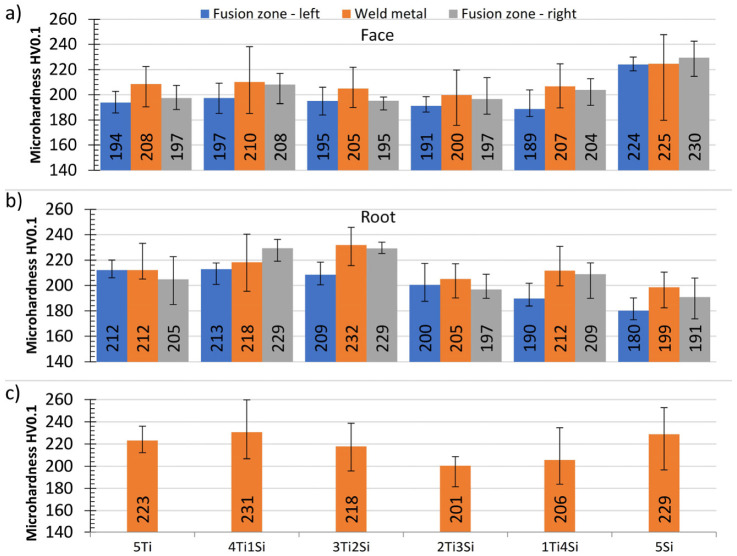
Average microhardness results for weld face horizontal measurements (**a**); weld root horizontal measurements (**b**); weld metal vertical measurements (**c**).

**Table 1 materials-13-04513-t001:** Chemical composition of the base metal wt.%.

C	Si	Mn	Cr	Ni	P	S	Fe
0.02	0.042	1.55	18.63	8.23	0.032	0.0004	balance

**Table 2 materials-13-04513-t002:** Mechanical properties of the 304L base metal and nominal base metal properties (ASTM A240/240M; EN ISO 10088-2).

Material Specifications	Tensile Properties				Impact Energy		
	Yield Strength Rp (MPa)	Ultimate Tensile Strength Rm (MPa)	Elongation A (%)	Reduction of Area Z (%)	Crack Formation (J)	Crack Propagation (J)	Impact Energy (J)
Base metal	474	697	57	60	107	166	273
304L nominal values	>170	>485	>40	–	–	–	>100

**Table 3 materials-13-04513-t003:** Weld metal widths.

Specimen		5Ti	4Ti1Si	3Ti2Si	2Ti3Si	1Ti4Si	5Si
Weld width (mm)	Face	10.5	9.9	7.7	8.7	14.0	6.7
	Root	9.4	8.5	8.5	8.6	10.8	8.8

**Table 4 materials-13-04513-t004:** Chemical compositions of the weld metals and 304L austenitic stainless steel in accordance with ASTM A240/A240M, wt.%.

Specimen	C	Si	Mn	Cr	Ni	P	S	Fe
5Ti	0.02	0.45	1.5	18.57	8.29	0.032	0.0006	balance
4Ti1Si	0.02	0.45	1.51	18.55	8.24	0.032	0.0006	balance
3Ti2Si	0.02	0.45	1.53	18.56	8.2	0.03	0.0007	balance
2Ti3Si	0.02	0.45	1.51	18.57	8.24	0.032	0.0007	balance
1Ti4Si	0.025	0.45	1.53	18.6	8.26	0.032	0.0006	balance
5Si	0.025	0.44	1.55	18.63	8.23	0.031	0.0004	balance
AISI 304L	0.02	≤0.045	≤1.55	18.63	8.23	0.032	0.0004	balance

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
