# Peer review of "Influence of Metallic Oxide Nanoparticles on the Mechanical Properties of an A-TIG Welded 304L Austenitic Stainless Steel"

_materials, 2020, doi:10.3390/ma13204513_

Round 1
Reviewer 1 Report
Comments and Suggestions for Authors
This study investigates the effect of nanoparticle-based flux on the microstructure and mechanical properties of the A-TIG welded 304L stainless steel. Overall, the manuscript is not well-written and the novelty is poor. The content of this manuscript is similar to that of the authors’ previous study (reference [12]). The article is lack of detail explanation, comprehensive comparison and the discussion is superficial. Most importantly, the English of this article is poor. I strongly recommend the manuscript polished by a native English speaker to make it more readable. Based on the above comments, I do not recommend publication of this manuscript. Some of my other comments are listed below:
Page 1: “Influence of Metallic Oxide Coatings on Mechanical Properties of A-TIG Welded 304 Austenitic Stainless Steel”
Please revise the title of this manuscript to “Influence of Metallic Oxide Nanoparticles on the Mechanical Properties of an A-TIG Welded 304L Austenitic Stainless Steel”.
Page 1: “This can be overcome by introducing a flux or a coating, applied before welding, which can reverse material flow and significantly increase penetration, acting as a welding catalyzer, even enabling the welding without consumable material. However, consumable also enables the conditioning of mechanical properties of welds.”
Please revise these sentences to make them more readable.
Page 1: “…, while bend testing, tensile and instrumented impact strengths were determined to evaluate mechanical properties of welds.”
Please delete the word “instrumented” and revise “impact strengths” to “impact toughness”.
Page 1: “Chemical compositions and weld mechanical properties were very similar in all specimens The highest tensile and impact properties were obtained with the specimen welded with the coating containing 1 % TiO2 and 4 % SiO2 nanoparticles. Although lower than the base metal, they were well within the nominal base metal mechanical properties.”
The meaning of these sentences is unclear. I strongly suggest revising the abstract by a native English speaker.
Page 1: “Keywords: oxide coating, activated TIG, mechanical properties”
Please revise “activated TIG” to “activated tungsten inert gas welding”. In addition, I suggest adding “stainless steel” or “304L stainless steel” as keywords. In addition, could you explain why choosing the term “oxide coating” instead of “oxide nanoparticles” as the keyword of this article?
Page 1: “Particular high-strength stainless steels also offers high strength-to-weight ratio, which makes them good alternative for aluminium alloys in aerospace applications. Their advantages also present their unique properties as hardness, strength, density, malleability, ductility at lower temperatures, elasticity, brittleness, conductivity, thermal expansion, good heat and fire resistance. Stainless steels are also easily fabricated using forming, machining, cutting and welding, which makes them good material for producing aerospace vehicles and components. Common applications of stainless steels in aerospace applications include springs, castings, tie rods, control cables, structural components like fasteners, flanges, airframes or fuselage structures and sheets, landing gear components, the wings, the cockpit, flash boilers, afterburners, exhaust components, engine parts and fuel tanks. Especially exhaust components, engine parts and fuel tanks are exposed to extremely corrosive materials and extremely high or low temperatures. Depending on the use and desired properties of components the following stainless steels are usually used in aerospace applications: 304, 316, 321, 410, 430, 13-8, 15-5, 17-4, 17-7 and 21-6-9 [1-3].”
This paragraph should be thoroughly revise since the target material of this manuscript is “austenitic stainless steel”, not “stainless steel”.
Page 2: “The application of activated flux tungsten inert gas welding (A-TIG) is an extremely attractive way of increasing the penetration and therefore, productivity, the major drawback of conventional TIG.”
Please revise this sentence to make it more readable. In addition, the authors do not explain the difference between the traditional TIG and A-TIG.
Page 2: “That means multi-pass is not needed if welding of stainless steels, titanium alloys, aluminium alloys, magnesium alloys and other steels is done [4,5].”
Please explain why.
Page 2: “Also, in A-TIG welding V-preparation is not needed, nor is the consumable material.”
I do not understand the meaning of this sentence. Please revise.
Page 2: “A-TIG was invented in the former USSR, more accurately, at the Paton Welding Institute in Kiev, present-day Ukraine [8].”
Please define “USSR”.
Page 2: “It consists usually of metallic oxides (SiO2, TiO2, Al2O3, NiO, MnO3, Cr2O3, CeO2, and V2O5) in alcohol or acetone solvents [5,9,10].”
All the numbers in the chemical compound should be shown in the right subscript position.
Page 2: “Several studies proved that smaller particle size possess a higher efficiency, since in arc heating, the thermal dissociation and decomposition of finer particles is quicker due to their higher specific surface area compared to micron particles [11-13].”
Do you mean “Several studies have proved that the nano-sized particles possess a high efficiency. This is because that the high specific surface area of the nano-sized particles resulted in a faster thermal dissociation and decomposition during arc heating compare to that of the micron-sized particles [11-13].”?
Page 2: “In conventional TIG welding, the molten metal flows from the center of the weld (low surface tension) towards the edges (high surface tension area), creating a relatively wide but at the same time, relatively shallow weld.”
Please revise this sentence to “In conventional TIG welding, the molten metal flows from the center of the weld (low surface tension) towards the edges (high surface tension area), creating a relatively wide but shallow weld.”
Page 2: “In A-TIG, this is reversed under the influence of oxygen from the flux, influencing a completely different weld shape: narrow and deep [14,15].”
Please revise this sentence to “In A-TIG welding, this is reversed due to the influence of oxygen from the flux, resulting a completely different weld shape: narrow and deep [14,15].”
Page 2: “It was found that A-TIG possesses a higher tensile strength compared to TIG with Inconel 82 filler wire, due to the presence of untempered martensite compared to austenite.”
What does the sentence “due to the presence of untempered martensite compared to austenite” mean?
Page 3: “In this study, the influence of nanoparticle-based fluxes on tensile properties, instrumented Charpy impact strength, microhardness, macro- and microstructures as well as the influence of activated flux composition on the chemical composition of the weld metal was examined. The obtained mechanical properties were compared to the base metal and procedures for determining the weld quality.”
The aim of this study is not clear. Please revise this paragraph to strengthen the novelty of this manuscript.
Page 3: “The welded specimens were designated in relation to the flux applied during welding: 5Ti (100% TiO2), 4Ti1Si (80% TiO2; 20% SiO2), 3Ti2Si (60% TiO2; 40% SiO2), 2Ti3Si (40% TiO2; 60% SiO2), 1Ti4Si (20% TiO2; 80% SiO2), and 5Si (100% SiO2).”
How much flux “ACTUALLY” applied during welding?
Page 3: “Microhardness was measured by Wilson Tukon 1102 (Uzwil, Switzerland) device with 1 kg loading, in base metal and weld metal as shown in Figure 2.”
Please revise “1 kg” to “1 kgf”.
Page 3: “Tensile and bend testing were done by the Schenck Hydropuls PSB 250 (Darmstadt, Germany) universal hydraulic testing machine, while Impact testing was performed on Testing Equipment IE JWT-450 (Jinan, China) instrumented Charpy pendulum tester.”
Detail of the tensile, bending and impact test should be indicated in the manuscript. For example, what is the strain rate used during the tensile test? What is the geometry of the specimen? Does the tensile, bending and impact test follow any standard? How many tests were carried out for each condition?
Page 4: “Although similar in shape and columnar morphology, typical of A-TIG weldments, there is a marked difference in width. In all specimens, the root is significantly wider than the face, as was obtained in [12], which is the result of the backing plate, that limited the excessive penetration.”
I suggest adding a table listing the width of the root and the face.
Page 4: “This indicates that the maximum penetration with applied parameters can be higher, that is, the present parameters can be applied to weld a thicker base metal of the same type.”
I do not understand what the authors try to present here. Please clarify.
Page 4: “The widest weld metal was obtained in specimen 1Ti4Si.”
Please explain why.
Page 4: “Microstructures obtained with the light microscope of the specimen 5Ti is shown in Figure 4.”
Are there any differences in microstructure between each testing condition?
Page 4: “SEM microstructure reveals the typical δ – ferrite morphology that occurs between austenitic dendrites.”
Where is this figure? Is it refer to Figure 5?
Page 6: “Tensile properties in all specimens are lower compared to the base metal, which can be the result of the replacement of rolled, strain hardened microstructure by cast dendritic microstructure, that is, the polygonal with dendritic austenite.”
What do you mean by stating “rolled, strain hardened microstructure”? Please clarify.
Page 6: “The highest tensile strength and contraction were obtained in the specimen 1Ti4Si, as the result of the widest weld metal and the widest space for deformation.”
Does that mean the mechanical properties of the weld metal is better than that of the base metal?
Page 6: “Non-metallic inclusions can be observed in Figure 8c) and d) at the bottom of some microvoids.”
Have you confirm the chemical composition of the non-metallic inclusions?
Page 6: “The highest tensile strength is the result of the highest amount of δ – ferrite in the austenitic matrix.”
Have you confirm the δ ferrite content of each specimen?
Page 6: “However, in this study, specimens obtained with SiO2 based flux exhibited a slightly higher tensile strength compared to TiO2, whitch is in contrast to the findings of Hdhibi et al. [22].”
Please explain why.
Page 7: “As the shape of the weld metal is completely different, with 1Ti4Si being noticeably wider and with a higher volume, it can be said that the size and shape of the weld metal do not represent a decisive parameter upon which impact energies depend.”
This statement contradicted to the previous statement “The highest tensile strength and contraction were obtained in the specimen 1Ti4Si, as the result of the widest weld metal and the widest space for deformation.”
Page 7: “Also, the presence of the less ductile δ – ferrite might explain why weld metals have a slightly lower impact energies compared to the base metal.”
This statement is questionable since the authors does not confirm the δ ferrite content of each specimen.
Page 7: “It can be seen that the curve shape is very similar, however, the overall impact energy of the base metal specimen is higher (blue accumulative curve).”
Please explain why.
Page 7: “In the majority of specimens, the hardness of the weld metal is higher compared to the hardness of fusion zones, probably due to a higher content of δ-ferrite and a more convenient ring-like versus linear morphology.”
There is no evidence in Figure 4 that could support this statement.
Page 9: “Although the differences are small, microhardness of specimens 2Ti3Si and 1Ti4Si are the lowest, which is interesting, since the tensile properties and impact energies of these specimens are the highest.”
There is no evidence in Figure 4 that could support this statement.
Figure 1:
The geometry of the specimens and the weld plate should be indicated in the figure. The location of the weld metal should be indicated in the figure as well. Furthermore, the end of the welding process (E) does not included in the figure.
Figure 3:
Scale bars should be indicated in the figure.
Figure 7:
What is the difference between Figure 7a and 7b? Please add more information in the figure caption.
Figure 8:
What is the difference between Figure 8a, 8b, 8c and 8d? Please add more information in the figure caption. In addition, please indicate the location of the non-metallic inclusions in the figure by using white arrows.
Figure 10:
What is the difference between the left-hand and right-hand side figure?
Table 2:
It seems that there is a huge difference in yield strength and ultimate tensile strength between the base metal and the nominal base metal. Please explain.
Author Response
Dear reviewer,
thank you for your review. We appreciate your valuable comments and suggestions which improve our paper. We considered all your comments and changed the paper accordingly.
authors

Reviewer 2 Report
The study reports an interesting investigation on understanding the effect of metallic oxide coating during TIG welding. Few items are to be addressed before publication. The manuscript can be improved by considering the following:
- Aerospace also looks in the lightweight properties of the materials. Authors have listed the advantages of steel. However, they need to highlight the lightweight aspects of aerospace components and how their approach addressing this issue.
- Please clarify the sentence ‘That means …’. Line 57. It seems the sentence is not complete or clearing revealing the underlying key points.
- Please report the standard followed for tensile specimen and testing.
- Figure 3 cross-sections do not seem they are butt welded (as mentioned in line 102-103). It is more like overlap weld. Authors need to clarify this point. Also, the nomenclature does not seem right. For example, it should be 100% TiO2 instead of 5% TiO2 and so on.
- The amount of nanoparticles also have an effect on the weld microstructure and strength. This is not discussed in the manuscript. Authors are requested to clarify this aspect. Probably, this is the reason for the weld width variation as shown in Figure 3. For example, 1TiO2+4SiO2 have the highest weld width. Also, put a scale within Figure 3.
- Authors are requested to provide joint cross-section without the application of flux. This will help to understand the effect of using flux.
- Authors need to clarify why 1Ti4Si has created widest weld. Please indicate if you have observed any trend in terms of average weld width with an increasing percentage of SiO2.
- Figure 11, HV0.1 indicated 100 gf rather 1kg as indicated in line 129. Please clarify.
Author Response

(The authors gave the same response as above.)

Reviewer 3 Report
The manuscript should be revised in the following parts:
Abstract / Introduction: the present manuscript correlates thematically with the results of one of your already published papers in Metals in 2019 (https://www.mdpi.com/2075-4701/9/5/567). This paper was once cited in text, however, there is no clear differentiation between what has already been done and what is new;
Materials:
Line 120-121: font style / size?
Figure 1 - There is no indication of where weld E-end and B-beginning are in the specimen.
Results: The interpretation of the results in Fig. 3 needs some further explanations. So in Fig. 3, fully penetrated welded seams are shown for the 10 mm plate thickness. It is also said that the width of the welds depends on the flux used. The used sheet metal is relatively thin compared to the weld width (sheet thickness = approx. weld width) and the heat accumulation at the bottom of the weld can cause the fusion zone to become wider. In this respect, a question about these results is how the results can be transferred to thicker sheets or how the 2D- and 3D-heat distribution mode is taken into account?
The scale in figure 3 is missing.
According to your previous investigations (s. link above), it was reported that the maximum weld depth and weld width was achieved when using the coating 2Si3Ti (increased TiO2 content). However, in the present investigations it is conclude that the widest weld metal with the largest potential for increased penetration was obtained in specimen 1Ti4Si (low TiO2 content). Why is there such a difference in the results?
It is therefore necessary to evaluate the present results in retrospect to the already published experiments.
- 3Ti1Si in Fig. 6?
Author Response

(The authors gave the same response as above.)

Round 2
Reviewer 2 Report
Few minor comments:
1) Please check the manuscript for typographical errors.
2) table 3 appeared twice. Please correct it.
3) Please concise the conclusions.
Author Response

(The authors gave the same response as above.)

Reviewer 3 Report
The manuscript was significantly improved. Nevertheless, there are still some further comments that should be considered. These are:
- there are some unclarities with the welding parameters used. In the Line 121 it was stated that a welding current of 250 A was used. In the Line 172 a welding current of 300 A is indicated. In contrast, a EWM Tetrix230 TIG welding machine was used, which can only supply 230 A. Please check and, if necessary, correct this information;
- check the table numbering. The number 3 appears twice in the text;
- Line 207-208: Fracture mode is fully ductile, with a notable necking, as shown in Figure 6. Do you mean Figure 7? Check the references to the figures.
- please unify the writing style: Figures 9a and b or Figure 8c) and d)
- please add a scale bar to the Figure 10.
Author Response

(The authors gave the same response as above.)
